Navigating the microbial community in the trachea-oropharynx of breast cancer patients with or without neoadjuvant chemotherapy (NAC) via endotracheal tube: has NAC caused any change?

Kim Hee Yeon 1
Kim Tae Hyun 1
Shin Jeong Hwan 2
Cho Kwangrae 3
Ha Heon-Kyun 4
Lee Anbok 4 ab-lee@hanmail.net
Kim Young Jin 5 khmclab@gmail.com
1 Department of Surgery, Busan Paik Hospital, Inje University , Busan , South Korea
2 Department of Laboratory Medicine and Paik Institute for Clinical Research, Inje University , Busan , South Korea
3 Department of Anesthesiology and Pain Medicine, Busan Paik Hospital, Inje University , Busan , South Korea
4 Department of Surgery, Chung-Ang University Gwangmyeong Hospital, Chung-Ang University College of Medicine, Chung-Ang University , Gyeonggi-do , South Korea
5 Department of Laboratory Medicine, Kyung Hee University College of Medicine, Kyung Hee University Medical Center , Seoul , South Korea
Chen Jun
Electronic publication date: 2023 Nov 23
Publication date: 2023
Volume: 11
Electronic Location ID: e16366
Received 2023 Apr 20; Accepted 2023 Oct 6
Copyright: © 2023 Kim et al.
Copyright year: 2023
Copyright holder: Kim et al.
License: This is an open access article distributed under the terms of the Creative Commons Attribution License, which permits unrestricted use, distribution, reproduction and adaptation in any medium and for any purpose provided that it is properly attributed. For attribution, the original author(s), title, publication source (PeerJ) and either DOI or URL of the article must be cited.
License URL: https://creativecommons.org/licenses/by/4.0/

Keywords: Breast cancer, Lower respiratory tract, Microbiome, Neoadjuvant chemotherapy

Funding: Chung-Ang University Research Grants in 2023 This research was supported by the Chung-Ang University Research Grants in 2023. The funders had no role in study design, data collection and analysis, decision to publish, or preparation of the manuscript.

==============================
Background

We compare the diversity and niche specificity of the microbiome in the trachea-oropharynx microbiome of malignant breast neoplasm with or without neoadjuvant chemotherapy (NAC) via NGS analysis.

Methods

We prospectively collected a total of 40 endotracheal tubes intubated from subjects, of whom 20 with NAC treated breast cancer (NAC group) and 20 with breast cancer without NAC (Non-NAC group). We generated 16S rRNA-based microbial profiles in IlluminaTM platform and alpha diversity indices were compared between groups. For the comparison of taxa abundance, linear discriminant analysis effect size method with Kruskal-Wallis test was used. The distribution of variables between the two groups was compared using the Mann-Whitney test. For beta diversity analysis, PERMANOVA was used.

Results

Among the diversity indices, the NAC group showed significantly lower Chao1, Inverse Simpson, and Shannon indices than the Non-NAC group. The three most frequent taxa of all two groups were Streptococcus (20.4%), followed by Veillonella (11.9%), and Prevorella (10.4%). This order was the same in NAC and non-NAC groups.

Conclusion

Here, we provide the first comparison data of the respiratory tract microbiome of breast cancer patients with or without NAC via NGS analysis. This study ultimately seeks to contribute to future studies on the lower respiratory tract in cancer patients with cytotoxic chemotherapy by establishing reliable control data.

Introduction

The human microbiome refers to the collection of all bacterial flora in the human body (Mammen & Sethi, 2016). Microbial communities represent the unique characteristics of specific body habitats, with their configuration and roles varying across and within organs (Costello et al., 2009). The respiratory tract is exposed to 7,000 L of air per day and 104–106/m3 microorganisms (Kumpitsch et al., 2019). Therefore, the microbiome niche in the respiratory tract is composed of diverse resident and transient bacterial communities (Dickson et al., 2016; Kumpitsch et al., 2019).

Advances in metagenomic analyses have made it possible to obtain complete or nearly complete genomic sequences from uncultured bacteria. Next-generation sequencing (NGS) is a large-scale method that can be used to comprehensively analyze the human microbiome community under various conditions (Garza & Dutilh, 2015). Recent studies performed using NGS-based microbiome analysis suggested that an imbalance in the normal flora, a state known as dysbiosis, is associated with a broad spectrum of diseases, from sepsis to cancer development (Chen et al., 2019).

Among living women diagnosed with cancer in the past 5 years, 7.8 million had breast cancer, making it the most prevalent cancer in 2020 (World Health Organization (WHO), 2020). Neoadjuvant chemotherapy (NAC) is currently the treatment of choice for locally advanced breast cancer (Rubovszky & Horvath, 2017). Recent NGS-based microbiome analyses revealed that the microbiota may be altered in patients with breast cancer receiving NAC. Moreover, increasing evidence suggests that chemotherapy can cause microbiota dysbiosis, which may harm the safety and efficacy of NAC (Iida et al., 2013; Liu et al., 2021; Nadeem et al., 2021). The gut microbiome has been linked to NAC in patients with breast cancer, which may affect the treatment response and post-NAC weight changes (Chapadgaonkar, Bajpai & Godbole, 2023; Wu et al., 2022). The oral microbiome has also been reported to undergo changes following anticancer chemotherapy in studies involving pediatric patients with various types of cancer (Proc et al., 2022). Therefore, regarding the significance of microbial composition in NAC, few studies have focused on the impact of breast cancer NAC on the respiratory microbiome (Chen et al., 2019). In this study, we compared the lower respiratory tract microbiome of patients with breast cancer who were treated or not treated with NAC.

Materials and Methods

Patients

Patients aged 19–64 years who underwent surgery or other procedures under general anesthesia using an endotracheal tube (ETT) for breast disease between March 2020 and October 2021 were included in the study. Patients with breast cancer were classified into two groups: the NAC group consisting of patients with breast cancer who were treated with NAC, and the non-NAC group consisting of patients with breast cancer who did not receive prior chemotherapy. Patients with the following conditions were excluded: those with a history of antimicrobial exposure within 3 months, those with respiratory lesions, those with current respiratory disease, smokers, patients with diabetes, patients with anemia (hemoglobin less than 8 g/dL), and patients with an intubation time of at least 8 h.

Specimen and clinical information collection

Before anesthesia, the patients were asked to gargle the oral cavity with 20 mL of distilled water for 30 s and place the solution in a 50 mL Falcon tube. After surgery and anesthesia, the ETT was removed from the bronchus and aseptically cut at 10 cm from the distal end. The distal part exposed to the lower airway was placed in a 50 mL Falcon tube. Collected samples were immediately transported to the laboratory and washed with DNA-free water using a vortex mixer. A negative control sample was prepared by placing an unused endotracheal tube in a Falcon tube and washing with DNA-free water. The washed solution was stored at −80 °C until analysis. The time from extubation to storage did not exceed 60 min.

The following medical records were reviewed: sex, age, height, body weight, hemoglobin level, current diagnosis, chemotherapy history, surgical procedure, surgery time, smoking history, diabetes, tuberculosis, and pneumonia.

Amplicon sequencing analysis and bioinformatics

Nucleic acids were extracted using a DNeasy PowerSoil Pro Kit (Qiagen, Hilden, Germany) according to the manufacturer’s instructions. Libraries were prepared according to the 16S Metagenomic Sequencing Library protocol (Illumina, San Diego, CA, USA). The V3-V4 region of the 16S rRNA gene was amplified using the V3-F: 5′-TCGTCGGCAGCGTCAGATGTGTATAAGAGACAGCCTACGGGNGGCWGCAG-3′ and V4-R: 5′- GTCTCGTGGGCTCGGAGATGTGTATAAGAGACAGGACTACHVGGGTATCTAATCC-3′ primers, which contained an overhang adapter. PCR products were purified using AMPure beads (Agencourt Bioscience, Beverly, MA, USA). Purified PCR products were used for sequencing library construction using Herculase II Fusion DNA Polymerase and Nextera XT Index Kit V2 (Illumina). The PCR products were qualified using an Agilent Technologies 2100 (Agilent Technologies, Santa Clara, CA, USA) with a TapeStation D1000 ScreenTape (Agilent Technologies). The libraries were sequenced on a MiSeq (Illumina) using a 600-cycle kit (2 × 300 bp), and paired-end FASTQ files were generated. Sequencing adapter and F/R primer sequences were removed using Cutadapt (v3.2) (Martin, 2011). Error correction steps, including quality filtering, denoising, merging, and chimeric sequence removal, were performed using R software (v4.0.3; The R Project for Statistical Computing, Vienna, Austria) with the DADA2 (v1.18.0) package, and amplicon sequence variants were prepared (Callahan et al., 2016). For comparative analysis of the microbial communities, normalization was performed by subsampling based on the number of reads of the sample with the minimum read number among all samples using QIIME (v1.9) (Caporaso et al., 2010). Each amplicon sequence variant was taxonomically assigned using BLAST+ (v.2.9.0) to the reference database (NCBI 16S rDNA database May 18, 2021) (Camacho et al., 2009). If the query coverage was less than 85% or identity of the matched area was less than 85%, the taxonomy was not assigned. The criteria and procedures for taxonomic assignment based on BLAST+ are described in the Supplemental file. QIIME (v.1.9) was used to calculate bacterial community diversity indices such as Shannon, inverse Simpson, and Chao1.

Statistical analysis

The linear discriminant analysis effect size method was used to compare taxa abundances between the two groups. The alpha value for the factorial Kruskal-Wallis test among classes was 0.05, and the threshold for the log-scale linear discriminant analysis score was two (Segata et al., 2011). The Mann-Whitney test was used to compare the distribution of alpha diversity indices between the two groups and to compare clinical data between groups using MedCalc version 11.5.1.0 (MedCalc Software, Ostend, Belgium). A p value of less than 0.05 was considered to indicate statistically significant results. For beta diversity analysis, Bray-Curtis dissimilarity was determined using PERMANOVA in the Vegan package in R (Adloff et al., 2003; Dixon, 2003).

Ethics statement

This study was approved by the Medical Ethics Committee of Inje University Busan Paik Hospital (IRB # BPIRB 2019-01-112-003), and written consent was obtained from all participants.

Results

Clinicopathological characteristics of patients and clinical specimens

Forty relevant specimens were collected from female patients in the NAC group (n = 20) and non-NAC group (n = 20). The median (interquartile range) age of the 40 subjects was 50 (44–55) years, and the age distribution did not significantly differ between the two groups (P = 0.096). The pre-chemotherapy clinical stage of the NAC group was significantly more advanced than that of the non-NAC group (P < 0.001). Neither group showed significant differences in clinicopathological characteristics, except for the pre-chemotherapy clinical stage.

In the NAC group, the NAC regimen for breast cancer included the following: a combination of pertuzumab, trastuzumab, docetaxel, and cyclophosphamide; a combination of doxorubicin and cyclophosphamide followed by docetaxel; and a concurrent combination of doxorubicin and docetaxel. The median (interquartile range) duration of chemotherapy and number of days from completion of cytotoxic chemotherapy to sample collection were 108 (105–120) and 22 (20–27) days, respectively.

The median operative time for the 40 patients was 177 min. The patient characteristics are summarized in Table 1.

Table 1 Clinical characteristics of the patients by study groups.

	NAC
(n = 20)	Non-NAC
(n = 20)	Total
(n = 40)	P	
Age (years)	53 (48–59)	46 (43–53)	50 (44–55)	0.096	
Body mass index	24.3 (21.8–26.9)	22.8 (21.3–24.8)	23.7 (21.4–25.5)	0.209	
Serum glucose (mg/dL)	105 (100–113)	105 (100–114)	105 (100–113)	0.756	
Intubated time (minutes)	175 (160–225)	182 (153–213)	177 (160–217)	0.850	
Duration of chemotherapy (days)	108 (105–120)	N/A			
Days from last chemotherapy	22 (20–27)	N/A			
Pre-chemotherapy clinical stage				<0.001***	
cI	0	12			
cII	9	8			
cIII	11	0			
Notes:

Abbreviations: NAC, Neoadjuvant chemotherapy; N/A, not applicable.

***P < 0.001 are considered as statistically significant.

Continuous variables are reported as median (interquartile range).

Genus level taxon by study group

After subsampling, the number of reads per sample was 46,125. A total of 169 genus-level taxa was found in all samples. Of these, 32 taxa showed a relative abundance of 2% or more in at least one sample (Fig. 1). The numbers of genus-level taxa were 145 and 123 in the NAC and non-NAC groups, respectively. The most frequent taxon (median, %) across all samples was Streptococcus (20.4%), followed by Veillonella (11.9%) and Prevotella (10.4%). This order was the same in the NAC (22.7%, 13.7%, and 9.4%, respectively) and non-NAC (17.9%, 10.9%, and 9.8%, respectively) groups. In linear discriminant analysis effect size analysis, Tannerella, Actinomyces, Centipea, Moryella, Streptobacillis, Leptotrichia, and Capnocytophaga were more abundant in the non-NAC group than in the NAC group (Fig. 2). Within each group, the taxon with the highest relative abundance in a single sample was Pseudomonas (92.7%) in sample N6 of the NAC group and Streptococcus (39.7%) in sample C2 of the non-NAC group. The results obtained for the negative control samples are summarized in the Supplemental File.

Figure 1 Genus level taxa found in endotracheal tube washed fluid of patients with breast cancer.

Taxa with a relative abundance of at least 2% in at least one sample were marked as individual taxa and others were pooled as others.

Figure 2 Microbiota comparison between Non-NAC group (red) and NAC (green) group.

Taxa shows significant differences in linear discriminant analysis score comparisons.

Comparison of diversity indices by study group

The NAC group showed significantly lower Chao1 (P = 0.047), inverse Simpson (P = 0.008), and Shannon indices (P = 0.005) than the non-NAC group (Table 2). There was also a difference (i.e., beta diversity) between the NAC and non-NAC groups, as determined using PERMANOVA (P = 0.047, permutations = 9,999). Complete data for the three diversity indices are presented in Table S1.

Table 2 Comparison of diversity indices between study groups.

Diversity indices	NAC	Non-NAC	p	
Chao1	188.7 (164.2–232.3)	227.8 (212.2–255.9)	0.047*	
Inverse Simpson	0.940 (0.927–0.956)	0.959 (0.945–0.966)	0.008**	
Shannon	5.227 (4.745–5.549)	5.670 (5.272–5.911)	0.005**	
Notes:

Abbreviations: NAC, Neoadjuvant chemotherapy.

*P < 0.05, **P < 0.01 are considered as statistically significant.

Continuous variables are reported as median (interquartile range).

Discussion

The traditional belief that “the normal lung is a sterile organ” persisted until recently, when the hypothesis was refuted by the results of NGS-based microbiome analysis (Dickson et al., 2016; Huffnagle, Dickson & Lukacs, 2017). The lower respiratory tract harbors various microorganisms that can be detected using conventional culture-dependent techniques. Early studies of the microbiota in the human respiratory tract were performed using specimens from bronchoalveolar lavage, bronchoscopic-protected specimen brushes, endotracheal aspirates, and sputum samples. Sputum is universally used for microbial analysis in laboratories but is typically not collected from healthy individuals without respiratory diseases (Dickson et al., 2014). In this study, we used ETT specimens for microbiome analysis. The ETT contacts the tracheal wall, thereby collecting microorganisms that dwell in the upper part of the lower respiratory tract. The ETT is then extubated via the oropharynx and mouth, which are the same tracts where sputum is normally extracted from the lungs. Therefore, we evaluated the ETT as a reliable specimen representing the tracheoropharyngeal microbiome. ETT has not been widely to analyze healthy respiratory microbiomes (Cho et al., 2021), mainly because intubation is more methodologically invasive than is collecting other specimens. Nonetheless, for patients undergoing surgery under general anesthesia, collecting ETT data is a simple and reliable means of normal lung microbiota analysis.

Previous studies demonstrated via NGS analysis that bacterial communities are abundant in the normal lower respiratory tract. These studies identified Prevotella, Veillonella, and Streptococcus as the main normal flora in the lungs (Dickson et al., 2015, 2014; Yu et al., 2016). In this study, microbiome diversity was lower in the NAC group than in the non-NAC group.

Healthy lungs possess a rich and diverse microbiome with a low bacterial population (Faner et al., 2017). Previous studies suggested that normal flora protect the lungs from external pathogens (Magalhaes et al., 2016), strengthen the immune system (Vatanen et al., 2016), and improve nutrient uptake. Our data suggest that NAC causes changes in the respiratory microbiota (Blanton et al., 2016). Previous studies showed that various lung diseases are associated with the dominance of a single taxon, a small group of taxa, or a loss of bacterial diversity (Tunney et al., 2013). Recent research expanded this knowledge by revealing dysbiosis of the lung microbiome in respiratory diseases such as chronic obstructive pulmonary disease, bronchiectasis, respiratory viral infections, asthma, and even lung cancer (Yagi et al., 2021).

Lung impairment is a well-known adverse effect of chemotherapy and can range from acute to subclinical modifications (Taghian et al., 2001; Wong & Evans, 2017). The mechanisms causing lung damage after chemotherapy have been suggested to be structural disruption of the airway system, immune defects, increased inflammatory cells and interleukins in the alveoli (Bhalla et al., 2000; Wong & Evans, 2017) or a systemic inflammatory response (Endo et al., 2004). Furthermore, Liu et al. (2021) suggested that alterations in the gut microbiota are related to pneumonia after chemotherapy. Based on the findings of previous studies, microbiome changes in the respiratory tract following chemotherapy may be associated with an increase in pneumonitis or pneumonia; further studies are needed to investigate the relationship.

Based on our findings, decreases in several taxa, such as Actinomyces, Centipea, and Moryella, were more prominent in the NAC group than in the non-NAC group. Therefore, further studies involving monitoring of respiratory microbiomes, including the above taxa, may improve the understanding of the microbial effects and side effects of NAC.

Compared to the non-NAC group, the NAC group had more advanced disease. A recent advance in NGS-based microbiome analysis included the discovery of a link between the human microbiome and diseases. Studies have emphasized that dysbiosis affects cancer development (Chen et al., 2019). Numerous studies have been conducted on the gut or breast tumor microbiome, and there is growing evidence suggesting that dysbiosis of the normal flora is associated with breast cancer stage. Xuan et al. (2014) indicated that a reduced bacterial load in the tumor tissue is associated with a higher breast cancer stage. Another study by Meng et al. stratified patients with breast cancer by histological grades and showed that grade III tissues had higher alpha diversity than grade I and II tissues (Meng et al., 2018). However, in our research conducted with ETT as the respiratory sample, we observed no significant difference in the diversity of indices according to the clinical stage in both NAC and non-NAC groups (Supplemental file). As this is the first investigation of the trachea-oropharynx microbiome in patients with breast cancer, whether the clinical stage of breast cancer affects the lower respiratory tract microbiome should be further examined. Although we could not clarify whether there was any biological association between breast cancer stage and respiratory microbial alterations, our results provide methodological guidance for further studies.

In the present study, only one sample showed a dominant taxon which occupied more than 50% of the total microbiota. The relative abundance of Pseudomonas in sample N6 was 93%. Nevertheless, no clinical symptoms or radiological findings of pneumonia were observed before or after surgery. As there were no signs or symptoms of a respiratory infection, sputum cultures were not performed. Thus, the presence of overdominant taxa in the 16S rRNA amplicon sequencing of the respiratory sample may not directly indicate the presence of bacterial pneumonia.

Conclusions

We provided initial evidence on the effect of NAC on the tracheo-pharyngeal microbiome of patients with breast cancer. Our results may be used as control data in lung microbiome research of breast neoplasm entities and chemotherapy.

Supplemental Information

Supplemental Information 1 The full data for the Chao1, Inverse Simpson, and Shannon diversity indices.

Abbreviations: NAC, Neoadjuvant chemotherapy

Click here for additional data file.

Supplemental Information 2 BLAST+ processing of multiple top hits, DADA2 parameters, and Negative control sample quality control result.

Click here for additional data file.

Supplemental Information 3 Raw data.

Click here for additional data file.

Supplemental Information 4 Clinical stage and diversity within groups.

Click here for additional data file.

Abbreviations

NAC Neoadjuvant chemotherapy

ETT Endotracheal tube

NGS Next-generation sequencing

Additional Information and Declarations

Competing Interests

Author Contributions

Human Ethics

Data Availability

The authors declare that they have no competing interests.

Hee Yeon Kim conceived and designed the experiments, performed the experiments, analyzed the data, prepared figures and/or tables, and approved the final draft.

Tae Hyun Kim performed the experiments, prepared figures and/or tables, and approved the final draft.

Jeong Hwan Shin performed the experiments, prepared figures and/or tables, and approved the final draft.

Kwangrae Cho performed the experiments, prepared figures and/or tables, and approved the final draft.

Heon-Kyun Ha performed the experiments, prepared figures and/or tables, and approved the final draft.

Anbok Lee conceived and designed the experiments, performed the experiments, analyzed the data, authored or reviewed drafts of the article, and approved the final draft.

Young Jin Kim conceived and designed the experiments, performed the experiments, analyzed the data, authored or reviewed drafts of the article, and approved the final draft.

The following information was supplied relating to ethical approvals (i.e., approving body and any reference numbers):

This study received approval and ethical clearance from the Medical Ethics Committee of the Inje University Busan Paik Hospital (IRB # BPIRB 2019-01-112-003) and written consent was obtained from every participant enrolled.

The following information was supplied regarding data availability:

The sequencing data are available at GenBank: PRJNA834839.

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
