# Peer review of "Navigating the microbial community in the trachea-oropharynx of breast cancer patients with or without neoadjuvant chemotherapy (NAC) via endotracheal tube: has NAC caused any change?"

_PeerJ, doi:10.7717/peerj.16366_

## Round 0.1 · original submission · Major Revisions

While the reviewers found the work interesting, they raised significant concerns. Please address the reviewers' comments in the revision.

Reviewer 1 ·

Basic reporting

The authors use clear English language with minor mistakes needing improvement. There is a name Prevorella instead of Prevotella.

Please introduce more update literature with an estimation of 16S rRNA method e.g: Proc P, Szczepańska J, Zarzycka B, et al. Evaluation of Changes to the Oral Microbiome Based on 16S rRNA Sequencing among Children Treated for Cancer. Cancers (Basel). 2021;14(1):7. Published 2021 Dec 21. doi:10.3390/cancers14010007.

The article is well structured, the figures and tables are clear.

The submission is "self-contained" and include results relevant to the hypothesis.

Experimental design

The paper presents original primary research in the field
Journal.

However, the research question should be formulated more directly as the research goal.

The investigation are performed to a high technical and ethical standards.

Methods are described with sufficient detail and information.

Validity of the findings

All underlying data have been provided; they are robust, statistically sound, &
controlled.
Conclusions are well stated, linked to original research question & limited to
supporting results.

Additional comments

The article addresses the important topic of changes in the microbiome during oncological treatment of breast cancer. It describes the use of a new method to assess changes in the lung microbiome.
In my opinion, the impact and novelty of the article are high.
The research questions should be more highlighted in the text.
Minor language mistakes should be improved.

Reviewer 2 ·

Basic reporting

Overall, the manuscript was fairly well written although it should be reviewed for some grammatical issues. Furthermore, I have provided some comments on missing background that should be addressed.

- Line 70: Missing a reference

- I wouldn’t call NGS surveys unbiased perhaps wide scale is a better term. Biases are introduced in various places including extraction, amplification sequencing etc.

- Line 78: I don’t think pestilential is the correct term for microbes enriched during dysbiosis states.

- In general, I felt that the introduction was missing information on previous work that has shown the various impacts of chemotherapy on the human microbiome as well as what we know about the human microbiome and breast cancer. (i.e. is it different from those without breast cancer etc.) This is better addressed in the discussion but should at least by highlighted within the introduction.

- Lines 158-159. What does it mean to show differences in pre-chemotherapy clinical stage. Please expand on this and talk about why this may or may not impact the results.

- Line 177 – typo

- In the discussion the authors suggest that changes in microbiome may be associated with lung outcomes during NAC but do not show this in their results.

- The authors provided raw data on patient information but failed to indicate whether the data sequencing data was available for re-analysis.

Experimental design

The authors indicate that this is the first study to examine the impact of NAC on the respiratory microbiome of breast cancer patients. However, within the introduction they could improve on the clarity of the motivation behind the paper and what future steps could be done with the results they have generated.

I did not see any issues with ethical standards in the paper, however, there was a number of technical issues with the largest being the lack of negative controls. Indeed, we know the sample area is of low biomass and as such is susceptible to continuation during sequencing and DNA extraction. At the very least the authors should have sequenced the DNA free water the sample was stored in as well as a sample tube that was not interested into a patient. Moreover, the methods section requires more details surrounding sequencing processing (i.e. what DADA2 parameters?).

Validity of the findings

The validity of their data is difficult to assess as the authors did not include any negative sequencing controls and did not provide any raw sequencing data to analysis. I have included below additional comments regarding the methods used for this paper.

- Assignment of taxonomy using BLAST against NCBI is an odd choice. How do they deal with multiple top hits etc.? This method is fairly old and much better methods with verifiable performance exist. It may be better to use the DADA2 taxonomic assignment tool.

- In abstract they say they used linear-mixed-effect models but then in the methods indicate the use of Lefse (which is not a tool that accepts random effects) I would suggest the additional use of other tools such as Maaslin2, ALDEx2 or ANCOM-BC.

- There beta diversity analysis indicates the use of PCA but then does not indicate whether any actual statistical test is used or if it was just visualized.

- It is unclear what MedCalc is doing as they indicated they already calculated diversity indices using QIIME V 1.9 and tested for differences using a Mann-Whitney test.

- Line 182-183 while PCA is a useful visualization it only represents 65% of the total variance making it not possible to determine if beta diversity differs by eye alone. Please test this with a rigorous statistical test such as measuring beta dispersion followed by PERMANOVA. Please do not show 3 dimensions in 2D space (only show first two PC2 or please plot additional PCs in separate plots). Finally, what type of beta diversity is being displayed here?

- Line 185-188 no statistics are given here just that they are “significantly different”. Please give test statistics.

---

## Round 0.2 · Minor Revisions

The reviewer is pretty satisfied with the revision. Please fix the minor issue before I recommend an accept.

Reviewer 2 ·

Basic reporting

The authors have significantly improved the reporting within their manuscript. I did not notice any major issues with grammar.

However, there seems to be some miscommunication within the abstract and methods section of this paper. In the current version of the manuscript the authors state within the abstract the use of linear-mixed-effect-models however, the authors do not use this statstical modelling method in their results. Instead they use lefse, kurskal-wallis, and PERMANOVA tests. The abstract should be fixed.

Experimental design

Authors have met all my concerns in this area.

Validity of the findings

I am statisified with the negative control results the authors produced. This has addressed my previous concerns with validity.

Additional comments

Thank you for taking the time to address all of my comments.

---

## Round 0.3 · accepted · Accept

Your manuscript is now in good shape for publication. Thanks for your contribution.